# Behavior change due to COVID-19 among dental academics—The theory of planned behavior: Stresses, worries, training, and pandemic severity

Nour Ammar[1], Nourhan M. Aly[1], Morenike O. Folayan[2], Yousef Khader[3], Jorma I. Virtanen[4], Ola B. Al-Batayneh[5], Simin Z. Mohebbi[6,7], Sameh Attia[8], Hans-Peter Howaldt[8], Sebastian Boettger[8], Diah A. Maharani[9], Anton Rahardjo[9], Imran Khan[10], Marwa Madi[11], Maher Rashwan[12,13], Verica Pavlic[14], Smiljka Cicmil[15], Youn-Hee Choi[16], Easter Joury[17], Jorge L. Castillo[18], Kanako Noritake[19], Anas Shamala[20], Gabriella Galluccio[21], Antonella Polimeni[21], Prathip Phantumvanit[22], Davide Mancino[23,24], Jin-Bom Kim[25], Maha M. Abdelsalam[26], Arheiam Arheiam[27], Mai A. Dama[28], Myat Nyan[29], Iyad Hussein[30], Mohammad M. Alkeshan[31], Ana P. Vukovic[32], Alfredo Iandolo[33], Arthur M. Kemoli[34], Maha El Tantawi[1]*

**1** Department of Pediatric Dentistry and Dental Public Health, Faculty of Dentistry, Alexandria University, Alexandria, Egypt, **2** Department of Child Dental Health, Obafemi Awolowo University, Ile-Ife, Nigeria, **3** Department of Public Health, Jordan University of Science and Technology, Irbid, Jordan, **4** Department of Clinical Dentistry, Faculty of Medicine, University of Bergen, Bergen, Norway, **5** Department of Preventive Dentistry, Faculty of Dentistry, Jordan University of Science and Technology, Irbid, Jordan, **6** Research Center for Caries Prevention, Dentistry Research Institute, Tehran University of Medical Sciences, Tehran, Iran, **7** Community Oral Health Department, School of Dentistry, Tehran University of Medical Sciences, Tehran, Iran, **8** Department of Cranio-Maxillofacial Surgery, Justus-Liebig University Giessen, Giessen, Germany, **9** Department of Preventive and Public Health Dentistry, Faculty of Dentistry, Universitas Indonesia, Depok, Indonesia, **10** Department of Oral & Maxillofacial Surgery, Faculty of Dentistry, Jamia Millia Islamia, New Delhi, India, **11** Department of Preventive Dental Sciences, College of Dentistry, Imam Abdulrahman Bin Faisal University, Dammam, Saudi Arabia, **12** Center for Oral Bioengineering, Barts and the London, School of Medicine and Dentistry, Queen Mary University of London, London, United Kingdom, **13** Department of Conservative Dentistry, Faculty of Dentistry, Alexandria University, Alexandria, Egypt, **14** Department of Periodontology and Oral Medicine, Medical Faculty University of Banja Luka, Banja Luka, Bosnia and Herzegovina, **15** Department of Oral Rehabilitation, Faculty of Medicine Foca, University of East Sarajevo, Foca, Bosnia and Herzegovina, **16** Department of Preventive Dentistry, School of Dentistry, Kyungpook National University, Daegu, Republic of Korea, **17** Centre for Dental Public Health and Primary Care, Institute of Dentistry, Barts and The London School of Medicine and Dentistry, Queen Mary University of London, London, United Kingdom, **18** Department of Dentistry for Children and Adolescents, Universidad Peruana Cayetano Heredia, Lima, Peru, **19** Oral Diagnosis and General Dentistry department, Dental Hospital, Tokyo Medical and Dental University, Tokyo, Japan, **20** Department of Preventive and Biomedical Science, Faculty of Dentistry, University of Science and Technology, Sanaa, Yemen, **21** Department of Oral and Maxillo Facial Sciences, Faculty of Medicine and Dentistry, Sapienza University of Rome, Rome, Italy, **22** Faculty of Dentistry, Thammasat University, Bangkok, Thailand, **23** Department of Endodontics and Conservative Dentistry, Faculty of Dental Medicine, University of Strasbourg, Strasbourg, France, **24** Department of Biomaterials and Bioengineering, INSERM UMR_S 1121, Strasbourg University, Strasbourg, France, **25** Department of Preventive and Community Dentistry, School of Dentistry, Pusan National University, Busan, Republic of Korea, **26** Department of Biomedical Dental Sciences, College of Dentistry, Imam Abdulrahman Bin Faisal University, Dammam, Saudi Arabia, **27** Department of Community and Preventive Dentistry, Faculty of Dentistry, University of Benghazi, Benghazi, Libya, **28** Orthodontics and Pediatric Dentistry Department, Faculty of Dentistry, Arab American University, Jenin, Palestine, **29** Department of Prosthodontics, University of Dental Medicine, Mandalay, Myanmar, **30** Department of Pediatric Dentistry, Mohammed Bin Rashid University of Medicine and Health Sciences, Dubai, United Arab Emirates, **31** Department of Pediatric Dentistry, Seoul National University Dental Hospital, Seoul, South Korea, **32** Department of Pediatric and Preventive Dentistry, School of Dental Medicine, University of Belgrade, Belgrade, Serbia, **33** Department of Endodontics, University of Salerno, Fisciano, Italy, **34** Department of Paediatric Dentistry & Orthodontics, School of Dental Sciences, University of Nairobi, Nairobi, Kenya



**Data Availability Statement:** The dataset is attached as an Excel file.

**Funding:** The authors received no specific funding for this work.

**Competing interests:** The authors have declared that no competing interests exist.

\* maha_tantawy@hotmail.com

# Abstract

## Objective

COVID-19 pandemic led to major life changes. We assessed the psychological impact of COVID-19 on dental academics globally and on changes in their behaviors.

## Methods

We invited dental academics to complete a cross-sectional, online survey from March to May 2020. The survey was based on the Theory of Planned Behavior (TPB). The survey collected data on participants' stress levels (using the Impact of Event Scale), attitude (fears, and worries because of COVID-19 extracted by Principal Component Analysis (PCA), perceived control (resulting from training on public health emergencies), norms (country-level COVID-19 fatality rate), and personal and professional backgrounds. We used multilevel regression models to assess the association between the study outcome variables (frequent handwashing and avoidance of crowded places) and explanatory variables (stress, attitude, perceived control and norms).

## Results

1862 academics from 28 countries participated in the survey (response rate = 11.3%). Of those, 53.4% were female, 32.9% were <46 years old and 9.9% had severe stress. PCA extracted three main factors: fear of infection, worries because of professional responsibilities, and worries because of restricted mobility. These factors had significant dose-dependent association with stress and were significantly associated with more frequent handwashing by dental academics (B = 0.56, 0.33, and 0.34) and avoiding crowded places (B = 0.55, 0.30, and 0.28). Low country fatality rates were significantly associated with more handwashing (B = -2.82) and avoiding crowded places (B = -6.61). Training on public health emergencies was not significantly associated with behavior change (B = -0.01 and -0.11).

## Conclusions

COVID-19 had a considerable psychological impact on dental academics. There was a direct, dose-dependent association between change in behaviors and worries but no association between these changes and training on public health emergencies. More change in behaviors was associated with lower country COVID-19 fatality rates. Fears and stresses were associated with greater adoption of preventive measures against the pandemic.

# Introduction

The novel coronavirus (COVID-19) pandemic has influenced all life aspects. The highly contagious nature of the disease and its fatal outcomes led to changes in lifestyle for many people

[1]. These lifestyle changes included social distancing, avoiding public places, more frequent hand washing, and wearing face masks in public [2]. These changes were sometimes associated with stress-inducing factors such as temporary unemployment, working from home, home-schooling of children, lack of physical contact with other family members, friends, and colleagues, and worrying that loved ones and important others may be infected [3, 4].

Researchers and academics also face the psychological impact of the COVID-19 pandemic. The sudden closure of schools mandated the adoption of e-learning technologies. This, coupled with the suspension of several research projects [5] and unemployment threats [6] may have created new stresses and added to already existing mental health conditions associated with work-life conflict [7–9].

Healthcare workers are at greater risk of COVID-19 infection than the general population because of their frequent contact with affected individuals. Dental professionals are especially vulnerable to infections during pandemics [8]. Dental academics—educators who train dental students—face high levels of stress resulting from heavy work overload and incompatibility between their ability to act and what is expected from them [10]. They are also liable to anxiety and fear attributed to the greater risk of infection during treatment provision in the dental office, especially during pandemics [11]. Mild anxiety helps people perform goal-directed tasks, is natural and may foster preventive behaviors during pandemics [12]. Severe anxiety, on the other hand, is associated with physical symptoms such as muscle tightening, hyperventilation, increased heart rate, sweating, trembling, fatigue, troubled sleeping, gastrointestinal disorders in addition to impaired cognitive skills [13]. Persistent severe anxiety may affect physical and mental well-being [14, 15].

The theory of planned behavior (TPB) posits that behaviors can be predicted by intentions to engage in these behaviors [16]. These intentions, in turn, are affected by the control that people perceive they have over their actions, by their attitude toward the behavior and whether they think it is useful, important or desirable, and by the norms they perceive to be prevailing around them. The TPB was previously used to explain dentists' behaviors including delivering prevention [17], reporting suspected violence [18], and managing drug users [19]. The change in behaviors among dentists due to the COVID-19 pandemic may be explained by the TPB including the control they perceive they have over avoiding infection by the disease because of previous training they received, worries because of the pandemic which may affect their attitudes and the importance they attach to adopting preventive behaviors, and the prevailing norms around them regarding the seriousness of the pandemic based on the fatalities it causes. Adopting preventive measures to avoid infection protects health care professionals, their families, patients, and the public. It is important to understand the factors associated with these behaviors and if they are impaired by the levels of stress these professionals have. A number of studies have assessed the psychological outcomes of the pandemic on health care workers [20–22] and the general population [23–26] but not on dental academics.

This study aimed to assess the psychological impact of the COVID-19 pandemic on dental academics and on changes in their behaviors as a result of the pandemic in several countries. The hypothesis of the study was that the TPB components are associated with change in dental academics' behaviors due to the pandemic.

## Methods

### Design

This was a cross-sectional study that used an online, multi-country survey to collect data from dental academics in several countries around the world between March 2020 and May 2020.

Ethical approvals for the study were obtained from Alexandria University, Egypt (IRB 00010556)-(IORG 0008839)/6-11-2016) and other institutions in participating countries.

## Participants and sampling

The study participants were a convenience sample of dental academics identified and contacted through their emails that were publicly available on the academics' institutional affiliation websites, in addition to direct personal invitation from local dental academics (collaborators) in the country, who reached them through professional social media groups or email lists. Participants were recruited and invited to complete the online survey if they were dental academics training and/or educating dental students in universities or institutions at the time of the study, regardless of their degree (BDS or higher) or their title (professor or lower) including clinical instructors, and if they consented to participate. Undergraduate and postgraduate students were not invited to participate, nor were private dentists.

Countries from which participants were recruited are listed in S1 Appendix. Sample size was based on assuming a 95% confidence level, 5% margin of error, and prevalence of severe stress = 10% [27, 28]. The calculated number of participants also ensured adequate power for Principal Component Analysis (PCA) which requires at least 100 participants [29].

## Study questionnaire

An anonymous, close-ended questionnaire was developed for the study. The questionnaire consisted of four sections; section 1 included the 15-item Impact of Event Scale (IES) [30, 31] which assessed post-traumatic responses to certain events- in this case, COVID-19. Its internal consistency and validity were previously demonstrated [30]. In the present study, its Cronbach alpha was 0.83 indicating high internal consistency. Items were scored on a 4-point Likert scale; 0 = not at all, 1 = rarely, 3 = sometimes, and 5 = often. Adding the scores of all items gave the total score which was categorized into subclinical, mild stress, moderate stress, and severe stress using cutoff points of 0–8, 9–25, 26–43, and 44+ [31]. Section 2 included 16 items assessing participants' attitudes toward the impact of the COVID-19. Participants indicated how much these items caused them worry on a scale from 1 (not worried at all) to 10 (extremely worried). Section 3 assessed participants' agreement with two statements describing change in behavior because of the COVID-19 pandemic (frequent handwashing and avoiding crowded places) on a scale from 1 (strongly disagree) to 10 (strongly agree). Section 4 was a 9 item close-ended questionnaire about participants' personal and professional background including sex, age, country, living arrangements, highest academic degree obtained, whether the participants coordinate courses, have clinical responsibilities, hold administrative positions, and whether they received training on public health emergencies, the full survey can be found in the S2 Appendix.

The questionnaire was uploaded to SurveyMonkey. Participants were asked to select only one response per question and they were allowed to make one submission. No IPs or emails were collected to ensure confidentiality. The questionnaire was preceded by a brief introduction explaining the purpose of the study, assuring participants of the confidentiality of their responses, and emphasizing that their participation was voluntary. After SurveyMonkey settings were modified, the survey was tested for face and content validity by five academics who were not involved in the study to ensure clarity and relevance of the questionnaire. The questionnaire was developed in English. In addition, two versions were prepared for use in Iran and Brazil where it was translated by collaborators/ dentists into Farsi and Portuguese followed by back translation to ensure accuracy.

### Data collection

Survey links were sent to collaborators for distribution to participants who received the links on their emails or social media groups. Reminders were sent two weeks after the first invitation email to encourage participation.

### Analysis

After the survey closure, the Excel sheets were downloaded, cleaned, and imported to SPSS version 23.0 for analysis (IBM Corp., Armonk, N.Y., USA). Frequencies, percentages, means, and standard deviations were calculated for descriptive statistics.

Prior to PCA, the suitability of data for this analysis was assessed. The Kaiser-Meyer-Olkin (KMO) measure of sampling adequacy was 0.91 which is above the recommended value of 0.6. The P-value of Bartlett's test of Sphericity [32] was statistically significant (P< 0.0001), supporting the use of PCA. Major attitude components were, therefore, extracted from the 16 items in section 2 of the survey. Extraction was based on eigenvalues >1. Varimax rotation with Kaiser normalization was used and loading coefficients < 0.4 were suppressed to facilitate interpretation of factor loading. Regression coefficients of the factors extracted from the PCA were saved to the dataset and used as explanatory variables for further analysis.

Two types of outcomes were assessed. The first was the stress levels based on the categories of IES. These were included in a multilevel ordinal logistic regression where the explanatory variables were the major worries/ attitudes derived from the PCA, the ratio between the number of COVID-19 deaths to cases per million at country level (fatality rate) [33] and whether the participant received training to manage public health emergencies. The model controlled for confounders (personal and professional background factors) which were introduced as fixed effects and country was included in the model as a random effect factor. The second set of outcomes was the scores indicating change in behaviors due to the COVID-19 pandemic (frequent handwashing and avoiding crowded places). These were included in two multilevel linear regression models with the same explanatory variables representing the TPB components (major attitude/ worries, fatality rate at country level, and receiving training) in addition to stress levels. These models also controlled for the confounders (personal and professional background variables) and included country as a random effect factor. Regression coefficients (and odds ratios for the ordinal logistic regression model) and 95% confidence intervals were calculated. Significance was set at 5%.

## Results

Responses were received from 1862 participants from 28 countries with an overall response rate = 11.3%. Almost half the participants (53.8%) were from Iran, USA, India, Germany and, Indonesia (Appendix 1). About 53.4% were females, 32.9% were >35–45 years old, 87.4% had clinical responsibilities and 52.9% had administrative positions. The mean (SD) fatality rate at country-level as of May 25[th], 2020 was 0.06 (0.04). Also, 9.9% had severe stress, 37.5% had moderate stress and 39.6% had mild stress Table 1.

Table 2 highlights the PCA and factor loadings for major worries and attitudes related to the COVID-19 pandemic. Three components explaining 67.3% of the variance were extracted by PCA from the 16 items with factor loadings ranging from 0.735 to 0.823. In the first component, seven items had loadings ≥ 0.735 and were related to fear of infection. In the second component, five items had loadings ≥ 0.737 and were related to worries from professional responsibilities. The last component included four items with factor loadings ≥ 0.754 and it was about worries from restricted mobility. The greatest fear of infection was that important others would get COVID-19 infection because of the participant (mean = 7.66). The greatest

**Table 1. Personal and professional background of dental academics and their levels of COVID-19- related stress (n = 1862).**

| Factors | | N (%) |
|---|---|---|
| Sex | Male | 869 (46.6) |
| | Female | 996 (53.4) |
| Age in years | 25–35 | 519 (27.8) |
| | >35–45 | 614 (32.9) |
| | >45–55 | 376 (20.2) |
| | >55–65 | 256 (13.7) |
| | >65 | 100 (5.4) |
| Living arrangements | Alone | 198 (10.6) |
| | With parents | 281 (15.1) |
| | With partner/ spouse | 1236 (66.3) |
| | Shared accommodation | 68 (3.6) |
| | Other | 82 (4.4) |
| Highest academic degree obtained | BDS | 337 (18.1) |
| | MSc | 619 (33.2) |
| | PhD | 909 (48.7) |
| Coordinates courses | No | 269 (14.4) |
| | Yes | 1596 (85.6) |
| Has clinical responsibilities | No | 235 (12.6) |
| | Yes | 1630 (87.4) |
| Has administrative position | No | 878 (47.1) |
| | Yes | 987 (52.9) |
| Received training for public health emergencies | No | 964 (51.7) |
| | Yes | 901 (48.3) |
| Stress levels | Subclinical | 242 (13.0) |
| | Mild | 739 (39.6) |
| | Moderate | 700 (37.5) |
| | Severe | 184 (9.9) |

worry about professional responsibilities was related to the required material during the pandemic (mean = 6.48). The greatest worry because of restricted mobility was caused by restricted mobility within the country (mean = 6.70).

Table 3 highlights the factors associated with stress levels among dental academics. Fear of infection, worries about professional responsibilities and restricted mobility, country-level fatality rate, and previous training on public health emergencies were significantly associated with severe, moderate, and mild stress ($P < 0.0001$). Fear of infection had a significant, direct, and dose-dependent association with stress with higher scores of fear associated in a gradient with mild (OR = 1.186), moderate (OR = 1.465), and severe stress (OR = 1.483).

Similarly, there was a significant, direct, and dose-dependent association between worries due to professional responsibilities and higher levels of stress: worries were associated in a gradient with mild (OR = 1.209), moderate (OR = 1.317) and severe stress (OR = 1.369). The direct, dose-dependent association between worries due to restricted mobility and stresses also followed a gradient with mild (OR = 1.010), moderate (OR = 1.302), and severe stress (OR = 1.379).

A stronger dose-dependent relationship was observed in the association between country-level fatality rate and severe (OR = 6.893), moderate (OR = 1.539), and mild stresses (OR = 0.947); higher fatality rate was associated with higher odds of severe and moderate stress

**Table 2. Principal component analysis and factor loadings for major worries and attitudes related to COVID-19 pandemic.**

| | Mean (SD) | Factor loadings | | |
| --- | --- | --- | --- | --- |
| | | Fear of infection | Professional responsibilities | Restricted mobility |
| Catching COVID-19 infection from a colleague | 5.83 (2.74) | 0.768 | | |
| Catching COVID-19 infection from a patient | 7.20 (2.80) | 0.786 | | |
| Catching COVID-19 infection from a student | 5.61 (2.96) | 0.777 | | |
| Catching COVID-19 infection from a source not related to work | 6.41 (2.61) | 0.735 | | |
| Important others getting infected with COVID-19 because of me | 7.66 (2.76) | 0.750 | | |
| Important others getting infected with COVID-19 because of another source | 7.64 (2.48) | 0.747 | | |
| Patients getting infected with COVID-19 | 7.15 (2.63) | 0.755 | | |
| Finishing open courses satisfactorily because of the COVID-19 outbreak | 6.18 (2.73) | | 0.774 | |
| Teaching students required material because of the COVID-19 outbreak | 6.48 (2.68) | | 0.823 | |
| Supporting students psychologically during the COVID-19 outbreak | 6.44 (2.67) | | 0.737 | |
| Managing online learning during the COVID-19 outbreak | 6.45 (2.75) | | 0.783 | |
| Finishing required reports/ assignments during the COVID-19 outbreak | 6.23 (2.76) | | 0.749 | |
| Restricted mobility in my country because of the COVID-19 outbreak | 6.70 (2.91) | | | 0.769 |
| Restricted mobility from/ to my country because of the COVID-19 outbreak | 6.13 (3.24) | | | 0.817 |
| Restricted mobility affecting sports and social activities because of COVID-19 | 6.33 (2.85) | | | 0.790 |
| Missing events important to my career because of the COVID-19 outbreak | 5.96 (2.94) | | | 0.754 |

KMO = 0.91, P value of Bartlett's test< 0.0001

but lower odds of mild stress. The association between stress levels and receiving training was U shaped; training was associated with higher odds of severe (OR = 1.040) and mild stress (OR = 1.084) and lower odds of moderate stress (OR = 0.971).

Multilevel ordinal logistic regression controlling for sex, age, living arrangements, highest academic degree obtained, course coordination, having clinical responsibilities, having administrative positions as fixed factors and country included as a random factor; 48.7% correctly classified. OR: odds ratio, CI: confidence interval, *: statistically significant at P< 0.05.

Table 4 shows the factors associated with behavior change as a result of the COVID-19 pandemic in multilevel linear regression analysis. Participants agreed that they avoided crowded places (mean = 8.14) and washed their hands frequently (mean = 8.06). Compared to subclinical stress, severe stress was significantly and directly associated with more frequent handwashing (B = 0.93) and avoiding crowded places (B = 0.62). Also, compared to subclinical stress, moderate and mild stresses were significantly associated with more frequent handwashing (B = 0.83 and B = 0.67) but had no significant association with avoiding crowded places. The dose-dependent associations between stress severity and change in each behavior followed a gradient with greater changes reported by participants with higher levels of stress.

**Table 3. Factors associated with stress levels among dental academics (n = 1862).**

| Stressors | OR (95% CI): vs subclinical stress | | |
| --- | --- | --- | --- |
| | Severe | Moderate | Mild |
| Fear of infection | 1.483 (1.481, 1.484)* | 1.465 (1.464, 1.466)* | 1.186 (1.184, 1.188)* |
| Worries about professional issues | 1.369 (1.368, 1.369)* | 1.317 (1.316, 1.318)* | 1.209 (1.208, 1.210)* |
| Worries about restricted mobility | 1.379 (1.378, 1.380)* | 1.302 (1.300, 1.305)* | 1.010 (1.008, 1.012)* |
| COVID-19 fatality rate | 6.893 (6.891, 6.893)* | 1.539 (1.539, 1.540)* | 0.947 (0.935, 0.939)* |
| Received training vs not | 1.040 (1.038, 1.042)* | 0.973 (0.971, 0.974)* | 1.084 (1.083, 1.086)* |

**Table 4. Association between change in behaviors due to COVID-19 and stresses, worries, COVID-19 fatality rate, and training among dental academics (n = 1862).**

| Factors | B (95% CI) | |
|---|---|---|
| | **Frequent handwashing** | **Avoiding crowded places** |
| Change in behavior scale: Mean (SD) | 8.06 (2.40) | 8.14 (2.41) |
| Severe vs subclinical stress | 0.93 (0.46, 1.40)* | 0.62 (0.15, 1.10)* |
| Moderate vs subclinical stress | 0.83 (0.48, 1.19)* | 0.33 (-0.03, 0.68) |
| Mild vs subclinical stress | 0.67 (0.33, 1.01)* | 0.26 (-0.08, 0.60) |
| Fear of infection | 0.56 (0.45, 0.67)* | 0.55 (0.44, 0.66)* |
| Worries about professional responsibilities | 0.33 (0.23, 0.44)* | 0.30 (0.19, 0.40)* |
| Worries about restricted mobility | 0.34 (0.24, 0.45)* | 0.28 (0.18, 0.39)* |
| COVID-19 fatality rate | -2.82 (-5.32, -0.32)* | -6.61 (-9.13, -4.08)* |
| Received training vs not | -0.01 (-0.21, 0.20) | -0.11 (-0.32, 0.10) |

Multilevel linear regression controlling for sex, age, living arrangements, highest academic degree obtained, course coordination, clinical responsibilities, having administrative positions as fixed factors and country as random factor,

B: regression coefficient, CI: confidence interval,

*: statistically significant at P< 0.05.

Greater fear of infection, worries about professional responsibilities and worries because of restricted mobility were associated with more frequent handwashing (B = 0.56, 0.33 and 0.34) and more avoidance of crowded places (B = 0.55, 0.30 and 0.28).

Higher COVID-19 fatality rates were associated with less frequent handwashing (B = -2.82) and less avoidance of crowded places (B = -6.61). The associations between receiving training and changes in the two behaviors were not statistically significant (P< 0.05).

## Discussion

The findings indicated that the COVID-19 pandemic was a stress inducer for dental academics, with approximately 10% having severe COVID-19-related traumatic stress. The main sources of stress were fear of contracting infection, restricted mobility due to the lockdown enforced in most countries to control the spread of the pandemic and worries because of professional responsibilities related to teaching and research. Measures taken by individuals to contain the infection included avoidance of crowded places and washing hands more frequently. Training on public health emergencies was significantly associated with stresses but not with change in behaviors due to the pandemic. A dose-dependent relationship existed between severity of stresses and worries related to fear of infection, teaching and research responsibilities and restricted mobility. A direct, dose-dependent relationship also existed between stress levels and change in behaviors due to the pandemic. Dose-dependent associations were suggested by Hill among the criteria supporting causality in observational studies when clinical trials cannot be conducted [34]. However, dose-dependent associations are not proof of causality on their own and the most important criterion of causality; time sequence where exposure precedes outcome, can only be ascertained in a longitudinal study. The study hypothesis was, thus, partly supported: not all components of the TPB were significantly associated with change in behaviors due to COVID-19.

One of the strengths of the study was the diversity of countries represented by the study participants. This enabled the study to generate data representing different educational systems and backgrounds thereby increasing the generalizability of the findings. Also, the study used validated tools with high internal consistency and/ or factor loadings. In addition, the

study captured the psychological impact of the pandemic at its early stages thus providing important and valuable information that can be used in designing support systems for dental academics.

The study, however, had some limitations. First, data were collected at different stages of the pandemic in various countries and this may have confounded the assessment of the level of stress. In addition, the study was cross-sectional and thus, cannot prove causality. Also, the response rate was low similar to previous research [35] and this may be attributed to the psychological impact of the pandemic with resulting possible underestimation of the level of stress reported in the study since those with higher levels of stress may be more likely to ignore the survey. The academics were selected in the present study using convenience sampling. Thus, strict statistical representativeness cannot be claimed. However, in the absence of a sampling framework including dental academics in educational institutions all over the world, random sampling may not be possible. Thus, the wide geographic coverage and range of professional attributes represented in the study indicate representativeness of a large segment of the dental academic community worldwide. The study highlights the psychological impact of the pandemic on dental educators who are critical stakeholders in the education and healthcare sectors. As countries pass through the first wave of the pandemic, more attention will need to be paid to the psychological impact of the pandemic on people's lives because of its possible effect on productivity, wellbeing, health, and quality of life [36].

We found a direct association between fears and worries and behavior changes in agreement with previous studies including British adults [37], a nationally representative sample of Americans [38] and lay persons from ten countries in Europe, America and Asia. [39] These studies reported an association between perceived risk of infection, fatality risk or negative emotions such as fear and anxiety and greater adoption of COVID-19 preventive behaviors such as hand hygiene and social distancing. Our findings and those of other studies suggest that fear may trigger a protective reaction through the adoption of preventive measures to reduce risks.

Lower COVID-19 fatality rates were observed in countries where dental academics reported more frequent handwashing and more avoidance of crowded places. Dental academics' COVID-19-related behaviors reflect the behaviors of the general populations. Risk reduction communication undertaken as part of the public health response might have led to the adoption of COVID-19 preventive measures resulting in lower rates of COVID-19 spread and fatality. However, the cross-sectional design of the study does not show time sequence and the direction of the relationship between fatality rates and adoption of preventive measures cannot be elucidated. Future longitudinal studies are needed to establish cause and effect and allow the disentanglement of these complex associations.

The present study showed a 10% prevalence of severe COVID-19-induced stress among dental academics; a higher level than the 7% reported among the general public in Wuhan, China [27] and 8.7% general anxiety reported among Italian dentists [28] and similar to the 11.5% among Israeli dentists and dental hygienists [40]. This indicates a need to provide support for dental academics' mental health. In addition, fear and anxiety among the educators may have a detrimental effect on dental students with long-lasting consequences on the profession. Few universities have instituted mental health support programs for their staff and students to cope with stresses even before the pandemic crisis [41] and this should assume greater importance as the duration of the pandemic becomes longer and its impact becomes greater.

In the present study, COVID-19 was associated with severe stresses because of restricted mobility caused by isolation and quarantine [42], as observed in past epidemics like the Ebola [43] and MERS [44]. The fear of transmitting infections to important others and loved ones was another COVID-19-related stress inducing factor observed in the present study similar to

that reported by dentists from 30 countries in a previous study [4], and it had a dose-dependent association similar to what was observed among Israeli dentists [40].

At the present time when the pandemic spreads and death toll rises, it was hoped that training would prepare dental academics to adopt preventive measures. Our results, however, showed no significant effect of previous training on changing behaviors. Training on public health emergencies was associated with less stress up to a certain level beyond which the higher awareness of risks brought about by training was associated with more rather than less stress. Thus, whether in relation to change in behavior or reducing stresses, training was not associated with greater perceived control. This may be attributed to the generic nature training the academics received which did not address the specific needs related to COVID-19 prevention. This implies that appropriate responses to COVID-19 will require specific and tailored training different from the standard training for public health emergencies. Targeted training programs developed by international organizations such as the World Health Organization may help provide the required skills to deal with this pandemic.

## Conclusion

The present study showed a considerable psychological impact of the COVID-19 pandemic on dental academics that was directly associated with fear of infection and worries because of professional responsibilities and restricted mobility. Changes in behaviors due to the pandemic and greater adoption of preventive measures were associated with stresses and worries in a direct and dose-dependent relationship but were not associated with training. Greater adoption of preventive measures was inversely related to COVID-19 fatality rates at country level.

## Supporting information

**S1 Appendix. Countries participating in the study and number of participants.**
(PDF)

**S2 Appendix. Survey for dental academics' stresses at the time of the COVID-19 outbreak (English, Portuguese, and Farsi versions).**
(DOCX)

**S1 Dataset.**
(XLSX)

## Acknowledgments

We are grateful to all the academics who kindly responded to the survey and answered our questions in these difficult times.

## Author Contributions

**Conceptualization:** Nour Ammar, Maha El Tantawi.

**Data curation:** Maha El Tantawi.

**Formal analysis:** Maha El Tantawi.

**Investigation:** Nour Ammar, Nourhan M. Aly, Morenike O. Folayan, Yousef Khader, Jorma I. Virtanen, Ola B. Al-Batayneh, Simin Z. Mohebbi, Sameh Attia, Hans-Peter Howaldt, Sebastian Boettger, Diah A. Maharani, Anton Rahardjo, Imran Khan, Marwa Madi, Maher Rashwan, Verica Pavlic, Smiljka Cicmil, Youn-Hee Choi, Easter Joury, Jorge L. Castillo, Kanako Noritake, Anas Shamala, Gabriella Galluccio, Antonella Polimeni, Prathip Phantumvanit,

Davide Mancino, Jin-Bom Kim, Maha M. Abdelsalam, Arheiam Arheiam, Mai A. Dama, Myat Nyan, Iyad Hussein, Mohammad M. Alkeshan, Ana P. Vukovic, Alfredo Iandolo, Arthur M. Kemoli.

**Methodology:** Nour Ammar, Nourhan M. Aly, Maha El Tantawi.

**Project administration:** Nour Ammar, Maha El Tantawi.

**Supervision:** Maha El Tantawi.

**Writing – original draft:** Nour Ammar, Nourhan M. Aly, Morenike O. Folayan, Maha El Tantawi.

**Writing – review & editing:** Nour Ammar, Nourhan M. Aly, Morenike O. Folayan, Yousef Khader, Jorma I. Virtanen, Ola B. Al-Batayneh, Simin Z. Mohebbi, Sameh Attia, Hans-Peter Howaldt, Sebastian Boettger, Diah A. Maharani, Anton Rahardjo, Imran Khan, Marwa Madi, Maher Rashwan, Verica Pavlic, Smiljka Cicmil, Youn-Hee Choi, Easter Joury, Jorge L. Castillo, Kanako Noritake, Anas Shamala, Gabriella Galluccio, Antonella Polimeni, Prathip Phantumvanit, Davide Mancino, Jin-Bom Kim, Maha M. Abdelsalam, Arheiam Arheiam, Mai A. Dama, Myat Nyan, Iyad Hussein, Mohammad M. Alkeshan, Ana P. Vukovic, Alfredo Iandolo, Arthur M. Kemoli, Maha El Tantawi.

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
