## [Decision Letter · Decision Letter 0]

19 Aug 2020

PONE-D-20-20341

Behavior change due to COVID-19 among dental academics - The theory of planned behavior: stresses, worries, training, and pandemic severity

PLOS ONE

Dear Dr. El Tantawi,

Thank you for submitting your manuscript to PLOS ONE. After careful consideration, we feel that it has merit but does not fully meet PLOS ONE’s publication criteria as it currently stands. Therefore, we invite you to submit a revised version of the manuscript that addresses the points raised during the review process.

We look forward to receiving your revised manuscript.

Kind regards,

Ratilal Lalloo

Academic Editor

PLOS ONE

Journal Requirements:

2. In your Methods section, please provide additional information about the participant recruitment method and the demographic details of your participants. Please ensure you have provided sufficient details to replicate the analyses such as: a) the recruitment date range (month and year), b) a description of any inclusion/exclusion criteria that were applied to participant recruitment, c) a table of relevant demographic details, d) a statement as to whether your sample can be considered representative of a larger population, e) a description of how participants were recruited, and f) descriptions of where participants were recruited and where the research took place.

3. Please include additional information regarding the survey or questionnaire used in the study and ensure that you have provided sufficient details that others could replicate the analyses. For instance, if you developed a questionnaire as part of this study and it is not under a copyright more restrictive than CC-BY, please include a copy, in both the original language and English, as Supporting Information. Moreover, please include more details on how the questionnaire was pre-tested, and transalted, and whether it was validated.

4. We noted in your submission details that a portion of your manuscript may have been presented or published elsewhere.

"We do not have a dual publication issue. We applied another survey to the same target group to addresss another research point. What is shared is only the sample profile. The dependent and independent variables are different. We are attaching the related paper which is submitted elsewhere for your review. "

Please clarify whether this publication was peer-reviewed and formally published. If this work was previously peer-reviewed and published, in the cover letter please provide the reason that this work does not constitute dual publication and should be included in the current manuscript.

Reviewers' comments:

Reviewer's Responses to Questions

**Comments to the Author**

1. Is the manuscript technically sound, and do the data support the conclusions?

Reviewer #1: Yes

Reviewer #2: Yes

Reviewer #3: Partly

Reviewer #4: Yes

Reviewer #5: Yes

2. Has the statistical analysis been performed appropriately and rigorously? 

Reviewer #1: Yes

Reviewer #2: Yes

Reviewer #3: No

Reviewer #4: Yes

Reviewer #5: Yes

3. Have the authors made all data underlying the findings in their manuscript fully available?

Reviewer #1: Yes

Reviewer #2: Yes

Reviewer #3: Yes

Reviewer #4: Yes

Reviewer #5: Yes

4. Is the manuscript presented in an intelligible fashion and written in standard English?

Reviewer #1: Yes

Reviewer #2: Yes

Reviewer #3: No

Reviewer #4: Yes

Reviewer #5: Yes

5. Review Comments to the Author

Reviewer #1: Dear Author, your manuscript is well written and relevant. please explain how your questionair was validated. It is not possible in my opinion to place such a long list of authors on one article of this kind; in my opinion you should select a maximum of 6 authors who gave the major contribute to this work and Aknowledge the others. You should add to the reference list an important reference specifically to this work and is the following:

Subjective Overload and Psychological Distress among Dentists during COVID-19.

Mijiritsky E, Hamama-Raz Y, Liu F, Datarkar AN, Mangani L, Caplan J, Shacham A, Kolerman R, Mijiritsky O, Ben-Ezra M, Shacham M.

Int J Environ Res Public Health. 2020 Jul 14;17(14):E5074. doi: 10.3390/ijerph17145074

Reviewer #2: I would like to take the chance and congratulate the authors for accomplishment of this work. As a dental academic I am happy with the way it was designed with likert scale questionnaire and conducted during the lockdown and limitation caused by COVID19 outbreak.

Reviewer #3: Typographical or grammatical: line 84 only one bracket needed; line 101 word psychological; line 145 suggest word 'these professionals' instead of 'the'; line 161 suggest adding a comma between 'websites' and 'in addition'; line 397 word psychological

Reference: line 211, in different format

Introduction: in general I would have liked to see a clear separation between academics who just teach, and those who also treat patients, or have any other additional duties (admin, etc). Reason for this is related to stress levels proven to increase with workload.

Line 129: Rutter et al. report higher levels of stress in healthcare workers due to a number of factors including low autonomy, work overload, and lack of congruence between power and responsibility. However, there is also evidence that taking on a teaching role in addition to their clinical role reduce job‐related stress. Also, there is no mention in this paper of pandemic-related issues. I suggest revising the text and separating the information pertaining to Rutter et al. and the information related to the other reference.

I would also like to read a bit more about a separation between mild and severe anxiety, and their impact on dentists' health. What is mild anxiety... and severe? What are the conditions that severe anxiety can cause to dentists' health... cardiovascular, digestive, etc? For example, line 129 'mild anxiety with symptoms of... is natural, and may foster preventative behaviors such as... However, severe and persistent anxiety as observed by symptoms of..., may cause the following conditions...'

Results paragraph - lines 239 to 246. Difficult to read as flow is impacted by % quoted... I suggest picking a small number of characteristics that generally describe the participant population, such as gender, age, highest education, clinical responsibilities and training for emergencies but not all % as these are already in the table. Clinical responsibilities will help separate those who just teach from those who teach and also treat patients (as above).

Stress scale used - I am curious as to why this scale was used. The study appears to be unclear in what stress they are discussing... personal or occupational?

Clarity with this could help determine which scale to use.

IES - there is a revised version of this scale, but it appears that the original scale was used. If this was to be used for a PTSD study I would be concerned, otherwise it should be ok.

However, the results for this scale do not seem right. This scale requires reporting mean scores for total score, and for intrusion, and avoidance subscales.

Also the test-retest reliability of this scale has been widely criticised (see for example Joseph, 2000).

I suggest the use of a pandemic-specific measure of stress such as the COVID-19 Pandemic Mental Health Questionnaire (CoPaQ), or COVID Stress Scales (CSS).

Final note - well done for studying this important topic, during these unprecedented times and the enormous impact of COVID-19. I wish you all the best!

Reviewer #4: This multinational cross-sectional study demonstrated a psychological impact of the COVID-19 pandemic on dental academics that was directly associated with fear of infection and worries related to professional responsibilities and restricted mobility. Training level was not significantly associated with behaviour change (according to theory of planned behaviour). Of interest, greater adoption of preventive measures (handwashing and crowd avoidance) was inversely related to COVID-19 national fatality rates. The study finds similar levels of anxiety as previously reported in the general populous and other dental professionals.

The study is well conducted and written, and appropriately highlights limitations of the research and its interpretation based on its cross-sectional and multinational nature.

I only have very minor comments:

1. Table 2 is cumbersome and would be better broken down into three separate tables based on factor loadings.

2. A literature search for similar psychological studies on other high-risk professional groups, e.g., ear, nose and throat specialists, should be conducted prior to any response to the editor, as the literature on COVID-19 is rapidly evolving. Similarly, any further comparisons that can be made with general populations may be beneficial.

Reviewer #5: This is a very well written, comprehensive analysis of the psychological impact of COVID-19 on dental academics. It is interesting to see how Theory of Planned Behavior has influenced academics across the world , irrespective of their differences. What is the direction the authors propose based on their study?

What other additional training would the academics require to overcome their fear/ stress of COVID-19 infection?

6. PLOS authors have the option to publish the peer review history of their article (what does this mean?). If published, this will include your full peer review and any attached files.

Reviewer #1: No

Reviewer #2: No

Reviewer #3: No

Reviewer #4: No

Reviewer #5: **Yes: **Dr. Parvati Iyer

---

## [Author Response · Author response to Decision Letter 0]

2 Sep 2020

Journal Requirements

• The ‘Revised Manuscript with Track Changes’ file has been revised and is compliant with the PLOS ONE style requirements, the files submitted have been renamed according to the instructions provided in the email for resubmitting the revised manuscript.

2. In your Methods section, please provide additional information about the participant recruitment method and the demographic details of your participants. Please ensure you have provided sufficient details to replicate the analyses such as:

a) the recruitment date range (month and year),

b) a description of any inclusion/exclusion criteria that were applied to participant recruitment,

c) a table of relevant demographic details,

d) a statement as to whether your sample can be considered representative of a larger population,

e) a description of how participants were recruited,

f) descriptions of where participants were recruited and where the research took place.

• We clarified the recruitment date (MS with track changes, L170-171) and the inclusion criteria (MS with track changes, L181-185). Table 1 includes the relevant demographic details. In MS with track changes, L176, we mentioned that we used convenience sampling. Later, in Discussion, MS with track changes, L368-373, we explain our point of view regarding representativeness. Details about recruitment of participants were added to Methods, MS with track changes, L169 -185 and 223-225.

3. Please include additional information regarding the survey or questionnaire used in the study and ensure that you have provided sufficient details that others could replicate the analyses. For instance, if you developed a questionnaire as part of this study and it is not under a copyright more restrictive than CC-BY, please include a copy, in both the original language and English, as Supporting Information. Moreover, please include more details on how the questionnaire was pre-tested, and translated, and whether it was validated.

• A copy of the questionnaire used has been submitted as the “S2 Appendix 2.pdf” supporting file, including the English, Farsi and Portuguese versions. Details on how the questionnaire was pre-tested and validated is highlighted in the “study questionnaire” subsection of Methods. Please check MS with track changes, L196 -197 and 216 -220.

4. We noted in your submission details that a portion of your manuscript may have been presented or published elsewhere.

"We do not have a dual publication issue. We applied another survey to the same target group to address another research point. What is shared is only the sample profile. The dependent and independent variables are different. We are attaching the related paper which is submitted elsewhere for your review. "

Please clarify whether this publication was peer-reviewed and formally published. If this work was previously peer-reviewed and published, in the cover letter please provide the reason that this work does not constitute dual publication and should be included in the current manuscript.

• The other publication is still under review and we have not yet received Reviewers’ comments. We targeted the same group (dental academics) using three different surveys with three different links. This means that one academic could have completed the three surveys or- for example- only one of them. The surveys were anonymous and therefore we cannot combine responses from the three surveys even if we wanted to do that. Thus, each survey must be included in a different paper. The present one in this paper was about stresses. The other one under review was about knowledge of various aspect of the pandemic: methods of transmission, diagnosis, manifestations, treatment and so on. Both topics are not related and cannot be combined they have different respondents/ samples of the same target group. The details of the other paper clarify this.

• We meant that the values for this particular variable were not in the table. They are already included in the data set we submitted as Excel file with the paper (column AZ: RatioDCases). We removed this part from the text to avoid misunderstanding.

Reviewers' comments

Reviewer #1

Dear Author, your manuscript is well written and relevant. please explain how your questionnaire was validated.

• Response: We thank the Reviewer for this comment. We highlighted in Methods section (MS with track changes, L196-197 and 215-220) how we assessed the internal consistency of relevant sections of the questionnaire and how we assessed its content validity by the help of some experts whose responses were not included in the final analysis.

It is not possible in my opinion to place such a long list of authors on one article of this kind; in my opinion you should select a maximum of 6 authors who gave the major contribute to this work and acknowledge the others.

• Response: We understand and thank the Reviewer for this concern. We followed the recommendations of the ICMJE (http://www.icmje.org/recommendations/browse/roles-and-responsibilities/defining-the-role-of-authors-and-contributors.html) for defining authorship. Because many countries were included in the study, we needed country-experts to be involved since they would be more able to represent their countries during sampling and to ensure the cultural appropriateness of the survey tool. Thus, the first criterion of authorship was fulfilled. During drafting and submission of the paper, the three other criteria were fulfilled. Selecting some authors and leaving out others for acknowledgement would have impinged on their rights. The number of authors, therefore, corresponds to the geographic coverage of the study and the input of collaborators. In addition, the journal did not require a maximum number of authors.

You should add to the reference list an important reference specifically to this work and is the following:

Subjective Overload and Psychological Distress among Dentists during COVID-19.

Mijiritsky E, Hamama-Raz Y, Liu F, Datarkar AN, Mangani L, Caplan J, Shacham A, Kolerman R, Mijiritsky O, Ben-Ezra M, Shacham M.

Int J Environ Res Public Health. 2020 Jul 14;17(14):E5074. doi: 10.3390/ijerph17145074

• Response: we thank the Reviewer for drawing our attention to the reference. We added it (reference #9).

Reviewer #2

I would like to take the chance and congratulate the authors for accomplishment of this work. As a dental academic I am happy with the way it was designed with likert scale questionnaire and conducted during the lockdown and limitation caused by COVID19 outbreak.

• Response: we are very thankful for the Reviewer’s kind feedback.

Reviewer #3

Typographical or grammatical: line 84 only one bracket needed;

• Response: we thank the Reviewer for catching this error. We removed the extra bracket.

Line 101 word psychological;

• Response: we corrected this word as instructed.

Line 145 suggest word 'these professionals' instead of 'the';

• Response: modified as suggested.

Line 161 suggest adding a comma between 'websites' and 'in addition';

• Response: we added the comma.

Line 397 word psychological

• Response: corrected as indicated.

Reference: line 211, in different format

• Response: We provided the information in between brackets because we considered the software as a product rather than a scientific publication that needs a reference to be cited. 

Introduction: in general I would have liked to see a clear separation between academics who just teach, and those who also treat patients, or have any other additional duties (admin, etc). Reason for this is related to stress levels proven to increase with workload.

• Response: we understand the Reviewer’s point of view and agree with it. We included item/s in the survey to assess patient care load of the academics, the number of courses they manage and whether they had administrative role. We controlled for all these factors in the analysis as confounders. We did not focus on them in Introduction since they were treated as confounders in addition to other confounders and because covering them all would have increased to the length of the Introduction and distracted the readers from the main aims of the study.

Line 129: Rutter et al. report higher levels of stress in healthcare workers due to a number of factors including low autonomy, work overload, and lack of congruence between power and responsibility. However, there is also evidence that taking on a teaching role in addition to their clinical role reduce job‐related stress. Also, there is no mention in this paper of pandemic-related issues. I suggest revising the text and separating the information pertaining to Rutter et al. and the information related to the other reference.

• Response: We rephrased to differentiate between the information supported by each of the two references following the Reviewer’s suggestion.

I would also like to read a bit more about a separation between mild and severe anxiety, and their impact on dentists' health. What is mild anxiety... and severe? What are the conditions that severe anxiety can cause to dentists' health... cardiovascular, digestive, etc? For example, line 129 'mild anxiety with symptoms of... is natural, and may foster preventative behaviors such as... However, severe and persistent anxiety as observed by symptoms of..., may cause the following conditions...'

• Response: Following the Reviewer’s suggestion, we clarified the difference between mild and severe anxiety, the manifestations of the latter and how they affect the person’s ability to function. We supported this by a new reference (#13) that was added to the references list.

Results paragraph - lines 239 to 246. Difficult to read as flow is impacted by % quoted... I suggest picking a small number of characteristics that generally describe the participant population, such as gender, age, highest education, clinical responsibilities and training for emergencies but not all % as these are already in the table. Clinical responsibilities will help separate those who just teach from those who teach and also treat patients (as above).

• Response: We edited to focus on the important variables that describe the profile of the participants based on the Reviewer’s suggestion.

Stress scale used - I am curious as to why this scale was used. The study appears to be unclear in what stress they are discussing... personal or occupational? Clarity with this could help determine which scale to use. IES - there is a revised version of this scale, but it appears that the original scale was used. If this was to be used for a PTSD study I would be concerned, otherwise it should be ok. However, the results for this scale do not seem right. This scale requires reporting mean scores for total score, and for intrusion, and avoidance subscales. Also the test-retest reliability of this scale has been widely criticised (see for example Joseph, 2000). I suggest the use of a pandemic-specific measure of stress such as the COVID-19 Pandemic Mental Health Questionnaire (CoPaQ), or COVID Stress Scales (CSS).

• Response: the scale is used to measure the stress that dental academics report because of the pandemic; their subjective stress. We did not use it to assess PTSD (https://link.springer.com/referenceworkentry/10.1007%2F978-94-007-0753-5_1377) and therefore opted to use the shorter 15-items original version to avoid respondent fatigue that is likely to occur because of the number of questions we used in the survey. This fatigue would have further reduced the response rate of the participants who are already burdened by their academic responsibilities at a difficult time. The IES in general was used in previous studies assessing the impact of COVID-19 on health care workers and/ or the general population. We followed the method of reporting the scale used in these studies where categories (mild to severe) were reported such as Shacham et al (https://pubmed.ncbi.nlm.nih.gov/32331401/), Bohlken et al (https://www.ncbi.nlm.nih.gov/pmc/articles/PMC7295275/), or total score was reported (Sun et al: https://pubmed.ncbi.nlm.nih.gov/32430086/, El-Zoghby et al: https://pubmed.ncbi.nlm.nih.gov/32468155/).

• We agree with the Reviewer that using pandemic-specific inventories would generate useful information about the psychologic impact of COVID-19. However, both scales that the Reviewer kindly suggested were not available at the time we planned our study; we began data collection in March 2020.

Final note - well done for studying this important topic, during these unprecedented times and the enormous impact of COVID-19. I wish you all the best!

• Response: we thank the Reviewer for the kind feedback and this appreciation of our work.

Reviewer #4

This multinational cross-sectional study demonstrated a psychological impact of the COVID-19 pandemic on dental academics that was directly associated with fear of infection and worries related to professional responsibilities and restricted mobility. Training level was not significantly associated with behaviour change (according to theory of planned behaviour). Of interest, greater adoption of preventive measures (handwashing and crowd avoidance) was inversely related to COVID-19 national fatality rates. The study finds similar levels of anxiety as previously reported in the general populous and other dental professionals.

The study is well conducted and written, and appropriately highlights limitations of the research and its interpretation based on its cross-sectional and multinational nature.

• Response: we appreciate the Reviewer’s positive feedback on our work.

I only have very minor comments:

1. Table 2 is cumbersome and would be better broken down into three separate tables based on factor loadings.

• Response: we understand the Reviewer’s concern. There are two reasons that we had for using one instead of three tables: 1) to avoid increasing the total number of tables to 6 which may be too many for readers to follow and 2) to follow the method of reporting the results of principal component analysis adopted in literature where the various factors to be reduced are displayed in one table showing the loadings into different components similar to, for example, these papers:

o https://www.scielo.br/scielo.php?script=sci_arttext&pid=S1415-790X2020000100472&lng=en&nrm=iso&tlng=en

o https://www.ncbi.nlm.nih.gov/pmc/articles/PMC7384865/

o https://www.ncbi.nlm.nih.gov/pmc/articles/PMC7395981/

o https://www.ncbi.nlm.nih.gov/pmc/articles/PMC7372913/

2. A literature search for similar psychological studies on other high-risk professional groups, e.g., ear, nose and throat specialists, should be conducted prior to any response to the editor, as the literature on COVID-19 is rapidly evolving. Similarly, any further comparisons that can be made with general populations may be beneficial.

• Response: After a review of literature based on the Reviewer’s suggestion, we added a number of references in Introduction assessing the psychological impact of the pandemic on health care workers and the general population.

Reviewer #5

This is a very well written, comprehensive analysis of the psychological impact of COVID-19 on dental academics. It is interesting to see how Theory of Planned Behavior has influenced academics across the world, irrespective of their differences. What is the direction the authors propose based on their study?

• Response: We highlighted in yellow the part of the Discussion where we emphasized the importance of paying attention to people’s psychologic health during the pandemic because of its importance for general wellbeing and the part where we indicated the responsibility of universities to stablish mental health support services for their teaching staff and students. 

What other additional training would the academics require to overcome their fear/ stress of COVID-19 infection?

• Response: We clarified this part at the end of the Discussion by indicating that programs designed specially to deal with COVID-19 -such as those developed by the WHO- may be helpful.

---

## [Decision Letter · Decision Letter 1]

17 Sep 2020

Behavior change due to COVID-19 among dental academics - The theory of planned behavior: stresses, worries, training, and pandemic severity

PONE-D-20-20341R1

Dear Dr. El Tantawi,

We’re pleased to inform you that your manuscript has been judged scientifically suitable for publication and will be formally accepted for publication once it meets all outstanding technical requirements.

Kind regards,

Ratilal Lalloo

Academic Editor

PLOS ONE

Additional Editor Comments (optional):

Reviewers' comments:

Reviewer's Responses to Questions

**Comments to the Author**

1. If the authors have adequately addressed your comments raised in a previous round of review and you feel that this manuscript is now acceptable for publication, you may indicate that here to bypass the “Comments to the Author” section, enter your conflict of interest statement in the “Confidential to Editor” section, and submit your "Accept" recommendation.

Reviewer #1: (No Response)

Reviewer #3: All comments have been addressed

Reviewer #4: All comments have been addressed

2. Is the manuscript technically sound, and do the data support the conclusions?

Reviewer #1: (No Response)

Reviewer #3: Yes

Reviewer #4: Yes

3. Has the statistical analysis been performed appropriately and rigorously? 

Reviewer #1: (No Response)

Reviewer #3: Yes

Reviewer #4: Yes

4. Have the authors made all data underlying the findings in their manuscript fully available?

Reviewer #1: (No Response)

Reviewer #3: Yes

Reviewer #4: Yes

5. Is the manuscript presented in an intelligible fashion and written in standard English?

Reviewer #1: (No Response)

Reviewer #3: Yes

Reviewer #4: Yes

6. Review Comments to the Author

Reviewer #1: (No Response)

Reviewer #3: All comments have been addressed and the manuscript is well written.

This topic is of great current importance.

Thank you for addressing the reviewers' comments.

Great work on this study!

Reviewer #4: The authors have satisfactorily addressed my comments on data presentation and the need for comparison to similar research in other health care workers and the general population.

7. PLOS authors have the option to publish the peer review history of their article (what does this mean?). If published, this will include your full peer review and any attached files.

Reviewer #1: **Yes: **Eitan Mijiritsky

Reviewer #3: No

Reviewer #4: No

---

## [Editor Report · Acceptance letter]

21 Sep 2020

PONE-D-20-20341R1

Behavior change due to COVID-19 among dental academics - The theory of planned behavior: stresses, worries, training, and pandemic severity

Dear Dr. El Tantawi:

I'm pleased to inform you that your manuscript has been deemed suitable for publication in PLOS ONE. Congratulations! Your manuscript is now with our production department.

Kind regards,

on behalf of

Dr. Ratilal Lalloo 

Academic Editor

PLOS ONE